# Verification of Nasogastric Tube Positioning Using Ultrasound by an Intensive Care Nurse: A Pilot Study

**DOI:** 10.3390/healthcare12161618

**Published:** 2024-08-14

**Authors:** María Robles-González, Oscar Arrogante, Juan Antonio Sánchez Giralt, Ismael Ortuño-Soriano, Ignacio Zaragoza-García

**Affiliations:** 1Intensive Care Unit, La Princesa University Hospital, 28006 Madrid, Spain; mroblesgonz@gmail.com (M.R.-G.); jasanchezgiralt@gmail.com (J.A.S.G.); 2Department of Nursing, Faculty of Nursing, Physiotherapy and Podology, Complutense University of Madrid, 28040 Madrid, Spain; iortunos@ucm.es (I.O.-S.); izaragoz@ucm.es (I.Z.-G.); 3Research Nursing Group of Instituto de Investigación Sanitaria Gregorio Marañón (IiSGM), 28007 Madrid, Spain; 4FIBHCSC, Instituto de Investigación Sanitaria Hospital Clínico San Carlos (IdISSC), 28040 Madrid, Spain; 5Invecuid Group, Instituto de Investigación Sanitaria Hospital 12 de Octubre (imas12), 28041 Madrid, Spain

**Keywords:** critical care nursing, enteral nutrition, gastric feeding tube, gastrointestinal intubation, intensive care unit, intubation gastrointestinal, reproducibility of results, ultrasonography

## Abstract

Placing a nasogastric tube (NGT) is a frequent nursing technique in intensive care units. The gold standard for its correct positioning is the chest X-ray due to its high sensitivity, but it represents a radiation source for critically ill patients. Our study aims to analyze whether the ultrasound performed by an intensive care nurse is a valid method to verify the NGT’s correct positioning and to evaluate the degree of interobserver agreement between this nurse and an intensive care physician in the NGT visualization using ultrasound. Its correct positioning was verified by direct visualization of the tube in the stomach and indirect visualization by injecting fluid and air through the tube (“dynamic fogging” technique). A total of 23 critically ill patients participated in the study. A sensitivity of 35% was achieved using direct visualization, increasing up to 85% using indirect visualization. The degree of interobserver agreement was 0.88. Therefore, the indirect visualization of the NGT by an intensive care nurse using ultrasound could be a valid method to check its positioning. However, the low sensitivity obtained by direct visualization suggests the need for further training of intensive care nurses in ultrasonography. According to the excellent degree of agreement obtained, ultrasound could be performed by both professionals.

## 1. Introduction

Placing a nasogastric tube (NGT) into a critically ill patient admitted to an Intensive Care Unit (ICU) is a frequent nursing technique. The main purpose of employing these tubes is usually for enteral nutrition (EN) administration [1]. In this sense, the latest guidelines related to critically ill patients’ nutrition from the European Society for Clinical Nutrition and Metabolism (ESPEN) [2] and the American Society of Parenteral and Enteral Nutrition (ASPEN) [3] recommend early EN initiation, during the first 48 h after the ICU admission whenever possible [4]. Other medical indications for NGT use are gastric emptying, abdominal tension reduction through stomach decompression, medication management, or gastric lavage procedures in poisoning cases [1].

The NGT introduction technique is commonly performed by an intensive care nurse caring for a critically ill patient. As this procedure is performed blindly, it is essential to verify the tube position once this nursing technique is completed [1]. Consequently, and without direct observation, possible complications related to NGT placement must be considered when performing this technique. A misplaced NGT may be located in the respiratory tract or esophagus. In the first case, complications involve the development of pneumothorax, pneumonia, pulmonary abscess, tracheal perforation, chemical pneumonitis, and respiratory distress syndrome, sometimes leading to the patient’s death [5,6]. In the second case, an insufficiently introduced tube significantly increases the risk of bronchopulmonary aspiration, resulting in respiratory complications [5,6].

Regarding the methods to check proper NGT position, the “gold standard” or reference method is the chest X-ray [7,8,9]. Most international guidelines strongly recommend this verification before using the tube [7,10,11]. These guidelines advise against epigastrium auscultation after air insufflation, which, despite being a widely disseminated method, does not allow distinguishing between respiratory, gastric, or intestinal locations [7,12,13]. It is also not advised to assume proper NGT position by merely inspecting gastric aspirate [6,7]. Other less sensitive methods to verify the NGT position include pH testing of gastric aspirate [7,14], capnography and colorimetric capnometry [7,15], or observing NGT introduction with an electromagnetic guide [7,16] or using a miniature camera embedded in the NGT distal end to aid in tube placement [7].

Although X-ray is the most precise method to verify the position of a recently placed NGT [11], it involves an undesirable effect, such as radiation accumulation in critically ill patients. For them, X-rays represent a strong ionizing source, potentially harmful to living tissues, with greater risk as the number of exposures increases [17]. Therefore, in recent years, the importance of using ultrasound as an alternative to X-ray in certain situations has been highlighted [18].

Given its non-invasive nature, ultrasound presents some advantages over X-ray: it is innocuous, painless, and well-tolerated by patients. In addition, if the proper equipment is available, it is less expensive than X-ray, offering immediate access, which allows obtaining real-time images and, therefore, reducing the time for results achievement [18].

Previous studies have demonstrated that ultrasound is an efficient method for verifying nasogastric tube placement [7]. Among the studies conducted to assess ultrasound as a method to verify NGT positioning, there are promising results in pediatric critical care [19], prehospital critical care [20], urgent and emergency care [21,22], hospitalized patients [23], and admitted patients in ICU [24,25,26], obtaining an ultrasound sensitivity varying from 85% to 100%. In this sense, it should be highlighted that physicians were the healthcare professionals who performed most of the ultrasounds in these studies. However, as NGT placement is considered a nursing technique [1], nurses should be logically responsible for both NGT placement and its position verification.

To date, sufficient evidence related to ultrasound use for verifying NGT positioning in intensive care nursing has not been found. In the latest systematic review on ultrasound for confirmation of NGT placement published in 2024, Tsujimoto et al. [27] included 22 studies; however, ultrasounds were performed by trained nurses in only 3 studies. Consequently, an assessment of the validity of ultrasound performed by an intensive care nurse when checking the NGT positioning in critically ill patients is needed. Therefore, the objectives of our study were to analyze whether the ultrasound performed by an intensive care nurse is a valid method to verify the NGT’s correct positioning and to evaluate the degree of interobserver agreement between a nurse and an intensive care physician in the NGT visualization using ultrasound.

## 2. Materials and Methods

### 2.1. Study Design

A cross-sectional and observational pilot study was carried out to validate a diagnostic test [28]. To control the methodological rigor and quality of this paper, the research and reporting methodology followed the STrengthening the Reporting of OBservational studies in Epidemiology (STROBE) [29] criteria for cross-sectional studies.

### 2.2. Study Setting

The study was conducted between February and June 2023 in the ICU of the tertiary-level La Princesa University Hospital in Madrid (Spain), which comprises 20 beds and attends mainly respiratory, cardiac, neurological, and post-surgical patients.

### 2.3. Participants

The selection criteria for the participating patients were critically ill adult patients with an NGT introduced by an intensive care nurse following a medical indication, admitted during the study period, including mechanically ventilated or not. Regarding the exclusion criteria, the patients excluded were those who: (1) did not provide their written informed consent; (2) presented contraindications for blind NGT introduction; and (3) had a Body Mass Index (BMI) >35 (obesity class 2). In this sense, a high BMI is associated with more adipose tissue and, consequently, poorer ultrasound visualization. As a poor-quality image hinders the result interpretation, this potential bias has been controlled following the recommendation of other authors [24,30].

The participating patients were included consecutively. Ultrasound verifications of the NGT positioning were performed successively, first by an intensive care nurse and then by an intensive care physician.

### 2.4. Variables

The patients’ variables collected were gender, age, weight, height, BMI, and reason for admission. We also considered whether they were intubated, tracheostomized or not. As the airway and trachea are supposed to be adequately obstructed in intubated and tracheostomized patients, placing an NGT in these patients often gives critical care nurses a false sense of security. However, NGT tube misplacement can lead to serious and life-threatening complications, so maintaining a safe practice procedure is essential in both intubated and tracheostomized patients [31,32].

The variables related to NGT were medical indication, number of lumens, type, and caliber. Furthermore, other variables related to our study aims were collected, such as NGT direct and indirect visualization (yes/no) by the intensive care nurse and physician using ultrasound and their degree of agreement in visualizing the NGT through both methods.

### 2.5. Data Collection

Once the critically ill patient or their representative had fulfilled and signed the written informed consent form, both the intensive care nurse and the intensive care physician performed an abdominal ultrasound on each patient independently and immediately, one after the other. It should be noted that this intensive care nurse had 10 years of experience attending critically ill patients, while the physician had 6 years of experience in intensive care. Both healthcare professionals took all the measurements to avoid possible variability bias in NGT visualizations. Before starting the study, the intensive care nurse attended three theoretical courses on ultrasound, of which a 100 h course was specifically related to abdominal ultrasound. Additionally, she was trained during four training sessions in ultrasound, of which two were specifically focused on abdominal ultrasound. These 3 h practical training sessions were provided by the radiology service of the participating hospital; during these, a radiologist trained the intensive care nurse before achieving adequate competency in abdominal ultrasound after 20 ultrasounds. On the other hand, the intensive care physician had received training in ultrasound during his medical education and routinely performed ultrasound in clinical procedures in critically ill patients.

The data were collected in a database created by the research team, where all the study variables were detailed.

### 2.6. Technique to Perform the Ultrasound

The ultrasound available in the UCI (Philips CX50™) was used in this study. A 3–5 MHz convex probe was applied, and an exploration routine with cross-sectional, longitudinal, and oblique cuts was followed, starting in the epigastrium. Initially, an attempt was performed through NGT direct visualization, whose ultrasound image appeared as a simple or double hyperechogenic line. Subsequently, another attempt was performed through NGT indirect visualization by injecting (as long as the NGT was not a newly introduced tube) a mix of 1 cm^3^ water and 30 cm^3^ air through the tube to verify indirect signs indicating that it was in the stomach. These signs consisted of the appearance of a moving mist with small hyperechogenic spots corresponding to air bubbles (“dynamic fogging” technique), which is used as an acceptance criterion for an NGT correctly placed in the stomach [23,25,33].

After performing NGT direct and indirect visualization using ultrasound, a chest X-ray was performed to verify proper NGT placement. Its position was checked and confirmed by another intensive care physician different from the one that had performed the ultrasound.

### 2.7. Data Analysis

The statistical analysis was performed using the IBM SPSS Statistics^TM^ software, version 25.0 for Windows (IBM Corp., Armonk, NY, USA). Mean values, standard deviations, frequencies, and percentages were calculated for the descriptive analysis.

To analyze the diagnostic precision of ultrasound as a valid method to verify the NGT positioning, its sensitivity and specificity were calculated, as well as positive and negative predictive values (PPV and NPV) obtained from the comparison with X-ray. The corresponding formulas from diagnostic test validity studies were used for their calculation: PPV (quotient between true positive rate [sensitivity] and false positive rate [1-specificity]) and NPV (quotient between false negative rate [1-sensitivity] and true negative rate [specificity]) [28].

Finally, the interobserver agreement degree between the intensive care nurse and intensive care physician who performed ultrasounds was analyzed through Cohen’s Kappa index. In this sense, values below 0.4 indicate a low agreement, moderate agreement between 0.4 and 0.6, good agreement between 0.6 and 0.8, and excellent agreement above 0.8 [34].

### 2.8. Ethical Considerations

This study was approved by the Research and Ethics Committee of the participating hospital (registration number: 5155). Furthermore, it was conducted according to the ethical principles for medical research of the International Declaration of Helsinki and the UNESCO Universal Declaration on Human Rights. In addition, our study complies with the requirements of Spanish Law 14/2007 on Biomedical Research and Law 41/2002 on Patients’ Autonomy. Each critically ill patient, or his/her representative, received the patient’s information sheet with a detailed explanation of the study, including the written informed consent form, allowing them to ask questions or solve their doubts.

## 3. Results

A total of 23 critically ill adult patients participated in our study, 15 men (65.2%) and 8 women (34.8%) aged between 20 and 82 years (median = 65; interquartile range = 24). The response rate was 100%, with no refusals to participate. The main reason for admission was neurological (n = 12; 52.2%), specifically stroke (n = 8; 34.7%), followed by heart surgery (n = 4; 17.4%), septic shock (n = 2; 8.7%), cardiorespiratory arrest (n = 2; 8.7%), autolytic attempt (n = 2; 8.7%), and respiratory disease (n = 1; 4.4%). Nineteen patients (82.6%) were intubated and four (17.4%) were tracheostomized.

Regarding the characteristics of the NGT placed, their main medical indication was EN administration (n = 18; 78.3%), most were single-lumen tubes (n = 18; 78.3%), most of them were polyurethane feeding tubes (n = 13; 56.5%), following by polyvinyl chloride (PVC) Salem™ tube (n = 9; 39.1%) and PVC esophageal balloon (n = 1; 4.4%), and their caliber was mainly of 12 French gauge (n = 13; 56.5%), followed by 14 French gauge (n = 10; 43.5%). Table 1 shows all the characteristics of the participating critically ill patients and NGT placed.

Regarding the ultrasound discrimination capability to determine NGT presence or absence in the stomach, Table 2 shows the validity analysis of the correct positioning of the NGT using ultrasound performed by the intensive care nurse and physician and its comparison with the X-ray, including their corresponding sensitivity and specificity, PPV, and NPV values.

The X-ray confirmed proper NGT positioning in 20 cases (87.0%). Concerning the remaining cases, one NGT was found bent in the patient’s mouth, and two NGTs were in the esophagus. Consequently, these three NGTs were replaced.

As for NGT direct visualization through ultrasound, the intensive care nurse and physician reported that 7 NGTs (30.4%) were placed in the stomach, obtaining a sensitivity of 35.0% and specificity of 100%. However, through NGT indirect visualization using the “dynamic fogging” technique, the intensive care nurse verified that 17 NGTs (73.9%) were placed in the stomach, whereas the intensive care physician verified 18 NGTs (78.3%). Therefore, the sensitivity and specificity of the indirect visualization obtained by the intensive care nurse were 85.0% and 100%, respectively. These values were similar to values obtained by the intensive care physician (86.0% and 100% respectively) (Table 2). Figure 1 shows the ultrasound image of a 14-French gauge nasogastric tube placed in the stomach (the red arrow indicates two hyperechogenic parallel lines).

Finally, the agreement degree between the intensive care nurse and intensive care physician when observing the NGT in the stomach was calculated. Using a 5% significance level, the interobserver agreement degree reached a Cohen’s Kappa coefficient of 0.88, indicating an excellent degree of agreement between both professionals [34].

## 4. Discussion

Our study analyzed the validity of an ultrasound performed by an intensive care nurse as a method to verify the presence of an NGT in the stomach. The ultrasound sensitivity values were higher through indirect visualization than direct visualization (85% vs. 35%), obtaining the same specificity values in both methods (100% vs. 100%).

On one hand, for critically ill patients in which the NGT was not visualized in the stomach by the intensive care nurse using ultrasound and those who really did not have it there, the specificity was 100%. This result is coherent with the latest systematic review on ultrasound for confirmation of NGT placement published in 2024 [27]. On the other hand, for critically ill patients in which the NGT was visualized in the stomach using ultrasound and those who really had the tube in its proper position, the sensitivity increased from 35.0% through NGT direct visualization to 85.0% through indirect visualization using the “dynamic fogging” technique [23,25,33]. This result is congruent with the study conducted by Duan et al. [35], which concluded that ultrasound sensitivity increased considerably to 92.2% and 100.0% whether ultrasound is combined, introducing an air–water mix through the NGT. In this sense, Piton et al. [36] proposed introducing gastric aspirate through the NGT to visualize all the turbulences in the ultrasound image, allowing the verification of the proper NGT positioning.

Regarding the previous studies in which nurses performed ultrasounds, Tai et al. [21] obtained a sensitivity of 52.2% and 88.4% for NGT direct and indirect visualization, respectively. Mak and Tam [37] obtained higher sensitivity values, 92.4% and 95.4% for NGT direct and indirect visualization, respectively. However, they only included home-care patients, and the participating nurses were trained and supervised during 10 ultrasound assessments. Recently, Brotfain et al. [26] concluded that nurses verified proper NGT positioning in the stomach using ultrasound in 78.0% of patients. Although this study did not provide any valid data, nurses received theoretical-practical training, and a supervision process was carried out during 50 ultrasounds before initiating the study.

It should be noted that calculations of predictive values in our study provided information about ultrasound safety. A PPV value of 100% was found in both direct and indirect visualization of the NGT, as well as low NPV values (19% and 50% for direct and indirect visualization, respectively). This result is also congruent with the recent systematic review on ultrasound for confirmation of NGT placement conducted by Tsujimoto et al. [27]. However, it is also noteworthy that most of the NGTs used in our study were made of polyurethane and 12-French gauge. Except for a 14-French Salem NGT in a patient with abundant activated carbon in the stomach, the rest of the unobserved and well-positioned NGT corresponded to 12-French thin feeding tubes. On the contrary, Chenatia et al. [20] used NGT of 14 and 18-French gauge, obtaining a sensitivity of 98.3%, but showing that the larger the NGT caliber, the better its visualization.

Furthermore, the excellent interobserver agreement degree obtained between the intensive care nurse and intensive care physician should be highlighted (Cohen’s Kappa = 0.88). We believe this result may contribute to the safe use and management of ultrasound by intensive care nurses, including ultrasound in their routine clinical practice, provided they are trained and receive regulated education and qualification. In this sense, our results indicate that intensive care nurses trained in ultrasound could safely perform this procedure in their daily nursing practice. However, ultrasound training is not included in nursing curricula, unlike in medical education. Therefore, it is necessary to include a regulated qualification in ultrasound in nursing study plans to promote its safe use and management among intensive care nurses.

Concerning the implications for the clinical practice of our study, incorporating ultrasound, and specifically using NGT direct visualization, as a method to verify proper NGT positioning by intensive care nurses would be useful in avoiding patients’ exposure to radiation sources. In addition, we believe it may constitute a method to monitor NGT positioning, following the recommendations from the American Association of Critical-Care Nurses (AACCN) [38], which recommends checking NGT positioning every 4 h. This check would be complicated using X-ray owing to its undesirable radiation for critically ill patients. And even using other methods, such as gastric pH assessment, as it may hinder enteral nutrition administration in these patients. On the contrary, ultrasound represents a non-invasive method to verify NGT positioning and is less expensive than X-ray when the necessary equipment is available. Additionally, both performing and interpreting ultrasounds are faster than X-rays as they do not require moving any large equipment and an X-ray technician or an intensive care physician to later interpret the X-ray. Therefore, using ultrasound in ICUs may positively impact critically ill patients’ health and reduce health-related costs (material and human) and waiting time required to use an NGT, allowing its use as soon as possible. In this sense, it would be advisable to have not only radiopaque NGTs available in ICUs but also echogenic NGTs, which may improve their visualization.

Although the excellent interobserver agreement was obtained, the training differences between the participating intensive care nurse and physician were a study limitation. On the other hand, ultrasound is considered an operator-dependent technique [38] and, consequently, it could constitute a potential bias, limiting our results. Therefore, we need to confirm our results by conducting more studies in intensive care nurses more trained in ultrasound. Moreover, as all participating critically ill patients were intubated or tracheostomized, this potential bias should be controlled by including patients without these clinical situations in future research. In addition, we conducted a pilot study collecting a reduced sample, so future studies should recruit larger samples. In this sense, although our results cannot be generalized, we have described the study setting and methods in detail, allowing other researchers to apply the same methods in other contexts. Another study limitation is related to the technological features of the ultrasound equipment available in the participating ICU, as not all ultrasound systems offer the same image quality and, therefore, the images obtained and their interpretation may be influenced.

## 5. Conclusions

The NGT indirect visualization through ultrasound could constitute a valid method for intensive care nurses to verify its proper positioning and could even become an alternative method to X-ray. However, the low sensitivity obtained for direct visualization suggests the need for better training in ultrasound for both intensive care nurses and physicians and the creation of a regulated qualification in ultrasound, especially for intensive care nurses, including ultrasound training in the nursing study plans.

Although our results cannot be generalized as we conducted a pilot study in a small sample, the agreement degree between the data obtained by the intensive care nurse and the intensive care physician was excellent. Consequently, ultrasounds could be properly performed by both healthcare professionals, requiring future studies to confirm this hypothesis.

Finally, the non-invasive nature of ultrasound and its easy accessibility evidence the suitability of this technique to expand its use in the routine clinical practice of intensive care nurses.

## Figures and Tables

**Figure 1 healthcare-12-01618-f001:**
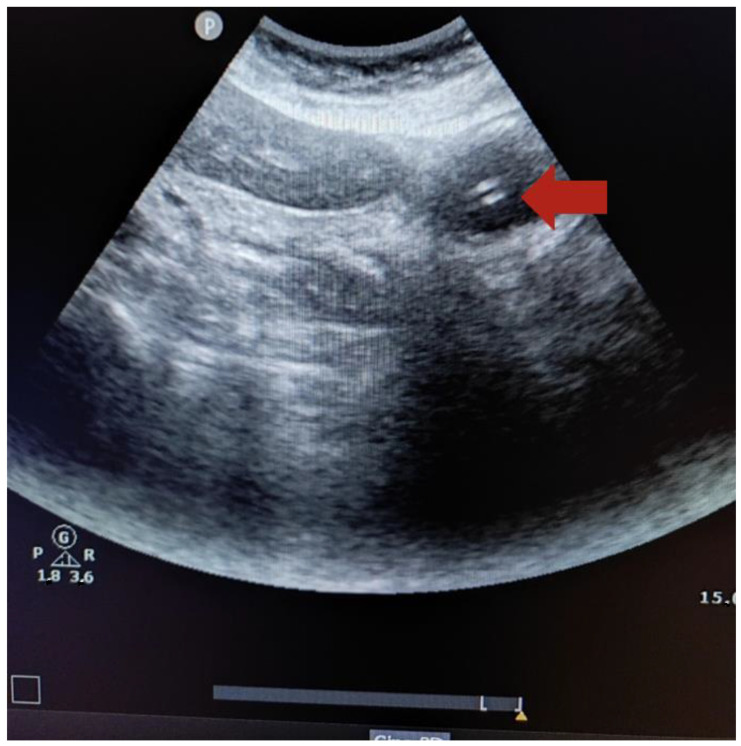
Ultrasound image of a 14-French gauge nasogastric tube placed in the stomach (the red arrow indicates two hyperechogenic parallel lines).

**Table 1 healthcare-12-01618-t001:** Characteristics of the participating critically ill patients and nasogastric tubes placed (n = 23).

**Characteristics of the participating critically ill patients**			**Median**	**Interquartile Range**
	Age (years)		65	24
	Weight (kg)		75	23
	Height (cm)		174	15
	Body mass index (kg/m^2^)		26.17	7.4
	Reason for admission		**Frequency (n)**	**Percentage (%)**
		Neurological disease	12	52.2%
		Heart surgery	4	17.4%
		Septic shock	2	8.7%
		Cardiorespiratory arrest	2	8.7%
		Autolytic attempt	2	8.7%
		Respiratory disease	1	4.4%
**Characteristics of the nasogastric tubes placed**	Medical indication	Enteral nutrition	18	78.3%
		Gastric emptying	4	17.4%
		Gastric lavage	1	4.4%
	Number of lumens	Single-lumen tube	18	78.3%
		Double-lumen tube	5	22.7%
	Type	Polyurethane feeding tube	13	56.5%
		PVC Salem™ tube	9	39.1%
		PVC esophageal balloon	1	4.4%
	Caliber	12-French gauge	13	56.5%
		14-French gauge	10	43.5%

Note: PVC—Polyvinyl chloride.

**Table 2 healthcare-12-01618-t002:** Validity analysis of the correct positioning of the nasogastric tube using ultrasound performed by the intensive care nurse and physician and its comparison with the X-ray.

	Ultrasound		X-ray		Total Ultrasoundn (%)	Sensitivity	Specificity	Positive Predictive Value	Negative Predictive Value
			Correct Positioning (n)	Incorrect Positioning (n)					
**Intensive care nurse**	Direct visualization	Yes	7	0	7 (30.4%)	35%	100%	100%	19%
		No	13	3	16 (69.6%)				
	Indirect visualization	Yes	17	0	17 (73.9%)	85%	100%	100%	50%
		No	3	3	6 (26.1)				
**Intensive care physician**	Direct visualization	Yes	7	0	7 (30.4%)	35%	100%	100%	19%
		No	13	3	16 (69.6%)				
	Indirect visualization	Yes	18	0	18 (78.3%)	86%	100%	100%	60%
		No	2	3	5 (21.7%)				

## Data Availability

The data presented in this study are available on request from the corresponding author.

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
