# Peer review of "Verification of Nasogastric Tube Positioning Using Ultrasound by an Intensive Care Nurse: A Pilot Study"

_healthcare, 2024, doi:10.3390/healthcare12161618_

Round 1

Reviewer 1 Report

Comments and Suggestions for Authors

Dear authors, you have submitted an innovative research article, focused in a very interesting and modern clinical topic such as the minimization of using X-rays and the increase of the use of U/S by well educated nurses. Please accept some comments for strengthening your research paper and maybe you can give more details.

Study Design

-Because of the very low population (n=23), please think if it is a pilot study and to refer it in the title/methods. It is not only a limiatation.

Methods

-How many of them (%) were already intubated? It is important, because the airway/trachea is already obstructed and usually there is not a false way of the NG tube. This gives more certainty to the placement of NG tube and the awareness where it is.

-You excluded the patients with a Body Mass Index (BMI) >35 (obesity class). Please clarify the rationale of this, why was necessary by a scientific opinion?

Results

If you have more statistics with the health personnel (e.g. age, years of experience in ICU, etc) it would be more interest to know who made the correct or false placement (if happened) or who made the indirect and direct test for the NGs. A statistic correlation would be interesting. Have you done this already?

Also please check…

in manuscript line 208. ….in the stomach by her using ultrasound….. what is ‘her’?

in line 234 …caliper… change to ‘caliber’

Best Regards

Comments on the Quality of English Language

Also, you need a minor polishing in english.

Author Response

Thank you for your kind words. We thank you for your positive assessment of our work and your suggestions. We appreciate the effort that you have made in reviewing our study. Working on your comments has undoubtedly helped us to improve our manuscript. All your suggestions and recommendations have been addressed in the responses below. Finally, we have highlighted all changes made in red throughout the revised manuscript.

Comments 1:

Study Design

-Because of the very low population (n=23), please think if it is a pilot study and to refer it in the title/methods. It is not only a limitation.

Response 1:

Thank you for your recommendation. We have modified the title, including ‘pilot study’ at the end. We have also included this issue in the Study Design subsection (p. 2, line 89). In addition, we have considered the small sample size as a limitation of our study at the end of the Discussion section (p. 8, lines 305-306), and included this issue in the Conclusions section (p. 8, lines 320-321).

Comments 2:

Methods

-How many of them (%) were already intubated? It is important, because the airway/trachea is already obstructed and usually there is not a false way of the NG tube. This gives more certainty to the placement of NG tube and the awareness where it is.

Response 2:

Thank you for your appreciation. We have included this issue in the Variables subsection (p. 3, lines 112-117).  As the airway and trachea are supposed to be adequately obstructed in intubated and tracheostomized patients, placing an NGT in these patients often gives critical care nurses a false sense of security. However, NGT tube misplacement can lead to serious and life-threatening complications, so maintaining a safe practice procedure is essential in both intubated and tracheostomized patients. We have also added 2 references supporting this sentence. In addition, we have also included the number of intubated and tracheostomized patients in the Results section (p. 4, lines 188-189). Finally, we have included this issue as a potential bias at the end of the Discussion section (p. 8, lines 304-306).

Comments 3:

Methods

-You excluded the patients with a Body Mass Index (BMI) >35 (obesity class). Please clarify the rationale of this, why was necessary by a scientific opinion?

Response 3:

Thank you for your suggestion. We have clarified the rationale of this exclusion criterion in the Participants subsection (p. 3, lines 103-106). A high BMI is associated with more adipose tissue and, consequently, poorer ultrasound visualization. As a poor-quality image hinders the result interpretation, this bias has been controlled following the recommendation of other authors. We have also added 2 references supporting this sentence.

Comments 4:

Results

If you have more statistics with the health personnel (e.g. age, years of experience in ICU, etc) it would be more interest to know who made the correct or false placement (if happened) or who made the indirect and direct test for the NGs. A statistic correlation would be interesting. Have you done this already?

Response 4:

Thank you for your appreciation. We must clarify all NGT visualizations were performed by the same intensive care nurse and physician to avoid possible variability bias in both direct and indirect visualizations. Consequently, we calculated the interobserver degree of agreement between these two observers. We have clarified this issue in the Data Collection subsection (p. 3, lines 129-130). Although we have included health professionals’ years of experience in intensive care to provide more information from these 2 observers (p. 3, lines 127-129), we did not calculate its statistical correlations as only two observers participated in our study. In addition, as we obtained an excellent degree of agreement between these 2 observers, possible differences in years of experience in intensive care were not considered.

Comments 5:

Also please check…

in manuscript line 208. ….in the stomach by her using ultrasound….. what is ‘her’?

in line 234 …caliper… change to ‘caliber’

Response 5:

Thank you for your suggestion. We have removed this pronoun to avoid any misunderstanding. We have also changed ‘caliper’ to ‘caliber’ throughout the manuscript.

Comments 6:

Also, you need a minor polishing in English.

Response 6:

Thank you for your recommendation. We must clarify that a specialized English reviewer has revised the English language again, correcting all grammatical and syntax errors. In this sense, we provide a certificate of translation services (please see the attachment).

Reviewer 2 Report

Comments and Suggestions for Authors

This is an interesting study to assess the nursing capacity to verify the correct position of the nasogastric tube using ultrasound, which requires a minor revision.

1) Introduction. The length of the introduction should be reduced. It is not necessary to refer to nutritional aspects.

2) The discussion should include what happens when the nasogastric tube is not observed. Is this because it is not correctly placed or because the nasogastric tube is not identified?

3) The authors should state both in limitations and in the conclusions the low sample size. Reference should be made in limitations that Cohen’s Kappa coefficient is 0.88 between intensive care nurse and intensive care physician due to a single patient.

4) The authors conclude the need for better training and regulated qualification in ultrasound for intensive care nurses. Intensive care physician should be included.

Author Response

Thank you for your kind words. We thank you for your positive assessment of our work and your suggestions. We appreciate the effort that you have made in reviewing our study. Working on your comments has undoubtedly helped us to improve our manuscript. All your suggestions and recommendations have been addressed in the responses below. Finally, we have highlighted all changes made in red throughout the revised manuscript.

Comments 1:

1) Introduction. The length of the introduction should be reduced. It is not necessary to refer to nutritional aspects.

Response 1:

Thanks for your recommendation. We have reduced the paragraph related to nutritional aspects in the Introduction section, highlighting only the importance of providing early enteral nutrition to critically ill patients according to international guidelines (p. 1, lines 30-35).

Comments 2:

2) The discussion should include what happens when the nasogastric tube is not observed. Is this because it is not correctly placed or because the nasogastric tube is not identified?

Response 2:

Thank you for your appreciation. This issue is mentioned in the Results section (p. 5, lines 213-215): ‘The X-ray confirmed proper NGT positioning in 20 cases (87.0%). Concerning the remaining cases, 1 NGT was found bent in the patient's mouth and 2 NGTs were in the esophagus. Consequently, these 3 NGTs were replaced’. Therefore, the NGT was misplaced when it was not observed using an X-ray to confirm its positioning, so it was replaced.

Comments 3:

3) The authors should state both in limitations and in the conclusions the low sample size. Reference should be made in limitations that Cohen’s Kappa coefficient is 0.88 between intensive care nurse and intensive care physician due to a single patient.

Response 3:

Thank you for your recommendation. We must clarify that we conducted a pilot study, so we have modified the title, including ‘pilot study’ at the end. We have also included this issue in the Study Design subsection (p. 2, line 89). In addition, we have considered the small sample size as a limitation of our study at the end of the Discussion section (p. 8, lines 305-306), and included this issue in the Conclusions section (p. 8, lines 320-321). On the other hand, we have included a sentence at the end of the Data Analysis subsection (p. 4, lines 170-172), indicating that Cohen’s Kappa values above 0.8 are very high. In addition, we have included a reference supporting this sentence. It should be noted that the intensive care nurse and the intensive care physician performed ultrasounds in all critically ill patients, so their degree of agreement was calculated considering all NGT direct and indirect visualizations (2 visualizations per patient performed by both intensive care professionals, a total of 92 visualizations).

Comments 4:

4) The authors conclude the need for better training and regulated qualification in ultrasound for intensive care nurses. Intensive care physician should be included.

Response 4:

Thank you for your suggestion. We have included this issue in the Discussion section (p. 8, lines 317-319), highlighting the need for better training in ultrasound for both intensive care nurses and physicians and the creation of a regulated qualification in ultrasound, especially for intensive care nurses, including ultrasound training in the nursing study plans.

Reviewer 3 Report

Comments and Suggestions for Authors

Thank you for the opportunity to review this manuscript titled “Verification of nasogastric tube positioning ultrasound by an intensive care nurse”.

The manuscript addresses an important topic of the nursing care in the intensive care settings, given the frequent use of NGT in ICU and the potential risks associated with its positioning.

I have some comments to improve your manuscript.

Introduction

The introduction is well refined, and highlights the importance of alternative methods to verify the correct positioning of NGT.

1.      Nevertheless, it is important to underline as first the main use of the NGT and then describe the other indications for its placement.

2.      I suggest you to revise the reference about the European Society for Clinical Nutrition and Metabolism (ESPEN) with the most up-to-date reference.

3.      I suggest you to rephrase at page 1 line 45-46 “Most complications…” to describe all the complications associated with the procedure.

4.      I suggest you to evaluate a recent review by Boeykens et al. (2023) to describe the methods used to verify the NGT position.

5.      At the final paragraph of the introduction (lines 78-81), it is necessary to include a reference.

Materials and method

The study design is clearly described, but the sample size (only 23 patients) is very small and this is a limit for the generalizability of the findings.

6.      It would be helpful to describe more detailed the selection criteria of the patients included in the study.

7.      I suggest you to describe in a more specific way the process of the nurses training sessions in ultrasound technique, in particularly about the whole training process and criteria for competency.

8.      Lines 120-121: explain and shortly describe the “ad-hoc database” used to collect data.

Results

9.  Regarding the characteristics of NGT, please indicated more details as lumen of the tubes (single/double), other materials besides polyurethane (if presents), and identify in detail some other reasons of admission (except by stroke).

10.  Line 183 “As for NGT indirect visualization…”, the term indirect is uncorrect. Please replace with direct.

Discussion

11.  I suggest you further exploring the section on the safe use and management of ultrasound by ICU nurses (lines 237-239).

13.  Please explain and discuss the limitations and the potential bias of the study.

Conclusion

The conclusion summarizes the findings of the study but it is necessary to underline that the results are not generalizable due to the sample size.

Author Response

Thank you for your kind words. We thank you for your positive assessment of our work and your suggestions. We appreciate the effort that you have made in reviewing our study. Working on your comments has undoubtedly helped us to improve our manuscript. All your suggestions and recommendations have been addressed in the responses below. Finally, we have highlighted all changes made in red throughout the revised manuscript.

Comments 1:

Introduction

The introduction is well refined, and highlights the importance of alternative methods to verify the correct positioning of NGT.

  1. Nevertheless, it is important to underline as first the main use of the NGT and then describe the other indications for its placement.

Response 1:

Thank you for your recommendation. We have underlined the main use of the NGT first and then described the other indications for its placement in the Introduction section (p. 1, lines 30-37).

Comments 2:

Introduction

  1. I suggest you to revise the reference about the European Society for Clinical Nutrition and Metabolism (ESPEN) with the most up-to-date reference.

Response 1:

Thank you for your appreciation. We have updated this reference:

  1. P.; Blaser, A.R.; Berger, M.M.; Calder, P.C.; Casaer, M.; Hiesmayr, M.; Mayer, K.; Montejo-Gonzalez, J.C.; Pichard, C.; Preiser, J.C.; Szczeklik, W.; van Zanten, A.R.H.; Bischoff, S.C. ESPEN practical and partially revised guideline: Clinical nutrition in the intensive care unit. Clin. Nutr. 2023, 42, 1671-1689. doi: 10.1016/j.clnu.2023.07.011.

Comments 3:

Introduction

  1. I suggest you to rephrase at page 1 line 45-46 “Most complications…” to describe all the complications associated with the procedure.

Response 3:

Thank you for your suggestion. We have rephrased this sentence, describing all the complications associated with a misplaced NGT in the respiratory tract or esophagus (p. 1-2, lines 42-47).

Comments 4:

Introduction

  1. I suggest you to evaluate a recent review by Boeykens et al. (2023) to describe the methods used to verify the NGT position.

Response 4:

Thank you for your suggestion. We have included this valuable narrative review to describe the methods used to verify the NGT position (p. 2, lines 48-57) and indicate that previous studies have demonstrated that ultrasound is an efficient method for verifying nasogastric tube placement (p. 2, lines 68-69).

Comments 5:

Introduction

  1. At the final paragraph of the introduction (lines 78-81), it is necessary to include a reference.

Response 5:

Thank you for your recommendation. We have included a reference to support these sentences for justifying the rationale of our study at the end of the Introduction section (p. 2, lines 78-80). In this sense, we have clarified that the latest systematic review on ultrasound for confirmation of NGT placement published in 2024 included 22 studies, however, ultrasounds were performed by trained nurses in only 3 studies. In addition, we have included this recent latest systematic review in the Discussion section (p. 7, lines 240-241; lines 263-265).

Tsujimoto, Y.; Kataoka, Y.; Banno, M.; Anan, K.; Shiroshita, A.; Jujo S. Ultrasonography for confirmation of gastric tube placement. Cochrane Database Syst. Rev. 2024, 7, CD012083. doi: 10.1002/14651858.

Comments 6:

Materials and method

The study design is clearly described, but the sample size (only 23 patients) is very small and this is a limit for the generalizability of the findings.

  1. It would be helpful to describe more detailed the selection criteria of the patients included in the study.

Response 6:

Thank you for your recommendation. We must clarify that we conducted a pilot study, so we have modified the title, including ‘pilot study’ at the end. We have also included this issue in the Study Design subsection (p. 2, line 89). In addition, we have considered the small sample size as a limitation of our study at the end of the Discussion section (p. 8, lines 305-306), and included this issue in the Conclusions section (p. 8, lines 320-321). In this sense, although our results cannot be generalized, we have described the study setting and methods in detail, allowing other researchers to apply the same methods in other contexts. On the other hand, we have described in detail the selection criteria of the patients included in the study in the Participants subsection (p. 2, lines 98-100).

Comments 7:

Materials and method

  1. I suggest you to describe in a more specific way the process of the nurses training sessions in ultrasound technique, in particularly about the whole training process and criteria for competency.

Response 7:

Thank you for your suggestion. We have described in detail the whole training process for abdominal ultrasound and the criteria for competency in the Data Collection (p. 3; lines 130-136). In this sense, we have indicated that the intensive care nurse attended 3 theoretical courses on ultrasound before starting the study, of which a 100-hour course was specifically related to abdominal ultrasound. Additionally, she was trained during 4 training sessions in ultrasound, of which 2 were specifically focused on abdominal ultrasound. These 3-hour practical training sessions were provided by the Radiology service of the participating hospital, during them a radiologist trained the intensive care nurse before achieving adequate competency in abdominal ultrasound after 20 ultrasounds.

Comments 8:

Materials and method

  1. Lines 120-121: explain and shortly describe the “ad-hoc database” used to collect data.

Response 8:

Thank you for your appreciation. We have replaced the ‘ad-hoc database’ with ‘a database created by the research team’ to avoid possible misunderstanding (p. 3, line 139).

Comments 9:

Results

  1. Regarding the characteristics of NGT, please indicated more details as lumen of the tubes (single/double), other materials besides polyurethane (if presents), and identify in detail some other reasons of admission (except by stroke).

Response 9:

Thank you for your recommendation. We have included more information about other reasons for admission and characteristics of NGT placed at the beginning of the Results section (p. 4, lines 185-195), including the number of lumens of the NGT and NGT materials. Table 1 (p. 5) shows all the characteristics of the participating critically ill patients and NGT placed, including the number of lumens of the NGT and NGT material.

Comments 10:

Results

  1. Line 183 “As for NGT indirect visualization…”, the term indirect is uncorrect. Please replace with direct.

Response 10:

Thank you for your appreciation. We have replaced it (p. 6, line 216).

Comments 11:

Discussion

  1. I suggest you further exploring the section on the safe use and management of ultrasound by ICU nurses (lines 237-239).

Response 11:

Thank you for your suggestion. We have clarified this issue in the Discussion section (p. 7; lines 275-280), indicating that our results show that intensive care nurses trained in ultrasound could safely perform this procedure in their daily nursing practice. However, ultrasound training is not included in nursing curricula, unlike medical education. Therefore, it is needed to include a regulated qualification in ultrasound in nursing study plans to promote its use and safe management among intensive care nurses.

Comments 12:

Discussion

  1. Please explain and discuss the limitations and the potential bias of the study.

Response 12:

Thank you for your appreciation. We have explained and discussed in detail the limitations and the potential biases of our study at the end of the Discussion section (p. 8; lines 299-312), proposing different solutions for future research.

Comments 13:

Conclusion

The conclusion summarizes the findings of the study but it is necessary to underline that the results are not generalizable due to the sample size.

Response 13:

Thank you for your recommendation. We have underlined that the results are not generalizable due to the sample size at the end of the Discussion section as a limitation of our study (p. 8; lines 307-309) and in the Conclusions section (p.8; lines 320-321).

Round 2

Reviewer 3 Report

Comments and Suggestions for Authors

Dear authors,

the quality of your paper has significantly improved following the suggestions provided during the review process. Thank you for your attention to the feedback and your efforts in revising the manuscript.